# Clinical Characteristics of Atrial Flutter and Its Response to Pharmacological Cardioversion with Amiodarone in Comparison to Atrial Fibrillation

**DOI:** 10.3390/jcm12134262

**Published:** 2023-06-25

**Authors:** Maciej T. Wybraniec, Kamil Górny, Kamil Jabłoński, Julia Jung, Kiryl Rabtsevich, Przemysław Szyszka, Fabian Wesołek, Karolina Bula, Małgorzata Cichoń, Wojciech Wróbel, Katarzyna Mizia-Stec

**Affiliations:** 1First Department of Cardiology, School of Medicine in Katowice, Medical University of Silesia, 47 Ziołowa St., 40-635 Katowice, Poland; 2Upper-Silesian Medical Center, 47 Ziołowa St., 40-635 Katowice, Poland

**Keywords:** atrial flutter, pharmacological cardioversion, amiodarone, atrial fibrillation

## Abstract

Background: Unlike atrial fibrillation (AF), atrial flutter (AFl) is thought to be relatively refractory to pharmacological cardioversion (PC), but the evidence is scarce. The aim of this study was to evaluate the clinical characteristics and efficacy of the PC of AFl with amiodarone in comparison to AF. Materials and methods: This retrospective study covered 727 patients with urgent consult for AF/AFl in a high-volume emergency department between 2015 and 2018. AFl was diagnosed in 222 (30.5%; median age: 68 (62; 75) years; 65.3% men). In a nested case-control study, 59 control patients with AF, matched in terms of age and sex with 60 AFl patients, were subject to PC with amiodarone. The primary endpoint was return of sinus rhythm confirmed using a 12-lead ECG. Results: The AFl population had a median CHA2DS2-VASc score of 3 (2; 4) and episode duration of 72 h (16; 120). In the AFl cohort, 36% of patients were initially subject to PC, 33.3% to electrical cardioversion (EC) and 40.5% to catheter ablation. In comparison to the AF group, the AFl patients required a longer hospitalization time, had a higher rate of EC (*p* < 0.001) and less frequent use of PC (*p* < 0.001) and, lower left ventricular ejection fraction (*p* < 0.001) and more pronounced cardiovascular risk factors. The efficacy of PC with amiodarone was significantly lower in AFl than AF group (39% vs. 65%, relative risk (RR) 0.60, *p* = 0.007). Conclusions: AFl patients shared a greater burden of comorbidities than AF patients, while the efficacy of PC in AFl was low. Patients should be initially managed with primary electrical cardioversion.

## 1. Introduction

Atrial flutter (AFl) is an organized supraventricular tachyarrhythmia with an atrial rate of 250–350/min, which is conditioned by the presence of an atrial macro-reentry circuit [1]. Although AFl may exist as a standalone arrhythmia, it frequently coexists with atrial fibrillation (AF) in the same patient. AF can develop in 10% of patients per year following successful catheter ablation for AFl [2,3]. More importantly, AF may shift into AFl following administration of Vaughan–Williams class I agents [4]. Accordingly, it is hard to delineate the true prognostic significance of AFl diagnosis adjusted to prevalence of AF.

AFl can be categorized into typical and atypical based on electrophysiological pathogenesis. The mechanism of typical AFl depends on right atrial cavotricuspid isthmus (CTI), while the atypical one does not and is often related with fibrosis of left atrium [1]. The electrocardiographic difference between atypical AFl and AF is frequently difficult to establish. Given the lack of mechanical function of atria, all patients diagnosed with AFl are classified using the CHA2DS2-VASc scale to evaluate the risk of thromboembolic incidents and to check if anticoagulant treatment is needed [5]. Still, this recommendation is mainly based on common sense and expert consensus, and it is not based on randomized data among the AFl population [6].

The clinical characteristics of patients in whom AFl is detected are different from patients with pure AF. Based on hitherto reports, patients with AFl have a greater prevalence of structural heart disease and heart failure reaching 50% of cases [7], as well as silent coronary artery disease [8]. In addition, AFl is particularly common in patients with chronic obstructive pulmonary disease [9] and after cardiac surgery [10]. Given the high efficacy of the procedure, patients with AFl should preferably be referred for catheter ablation, especially in cases of typical right atrial AFl [11,12]. The shortcomings of the health care system make it impossible to invasively manage all urgent patients presenting with AFl with tachyarrhythmia. Although acute rhythm management with pharmacological cardioversion in symptomatic episodes of arrhythmia is a viable option in AF, AFl is thought to be refractory to antiarrhythmic drug (AAD) administration [11]. Accordingly, current guidelines recommend upfront electrical cardioversion in atrial macro-reentry tachyarrhythmia [13,14]. Despite its efficacy in AFl patients, overdrive pacing is limited to patients with a cardiac implantable electronic device, as transesophageal stimulation is rarely applied in acute setting [15]. It is vital to note that, contrary to current guidelines [13], pharmacological cardioversion is frequently the first-line management of AFl. Given both the higher rate of structural heart disease in the AFl population [7] and the risk of 1:1 atrioventricular conduction in the case of class I drugs [16], class III agents seem to be the most suitable AAD class in this clinical scenario [17]. Ibutilide and dofetilide were documented to be particularly effective in converting AFl to sinus rhythm, but they are not universally available and are linked to a risk of significant proarrhythmia related with QTc prolongation [18,19]. Thus, in real-world conditions amiodarone is frequently initially administered in AFl patients [17], but its efficacy in acute rhythm management of AFl has not been diligently verified. Thus, the aim of the present study was to evaluate the clinical characteristics and management of patients with AFl and to assess the efficacy of pharmacological cardioversion of AFl with amiodarone in comparison to AF.

## 2. Methods

The study was carried out as a retrospective analysis of 727 patients consulted in the emergency department of the Upper-Silesian Medical Center in Katowice, Poland, between 2015 and 2018. The nested case-control study included 59 patients with AFl subject to pharmacological cardioversion with amiodarone matched in terms of age and sex with a control group of 60 patients with AF. Patients were treated in accordance with current guidelines and at the discretion of the attending physician. Acute rhythm control management comprised IV amiodarone dissolved in 5% glucose solution at a dose chosen by the attending physician (Cordarone^®^, Sanofi-Aventis, Bridgewater, NJ, USA). In general, the local protocol involves administration of 300 mg amiodarone dissolved in 50 mL of 5% glucose infused over 2 h; however, some patients received higher doses, as well as an initial bolus of 150 mg amiodarone. Electrical cardioversion was performed after a 12 h observation period according to the clinician’s judgement based on the chance of sinus rhythm restoration if pharmacological cardioversion failed to restore sinus rhythm. In patients with unsuccessful pharmacological cardioversion and no attempt at electrical cardioversion, a rate control strategy was introduced with delayed elective electrical cardioversion or ablation. Thus, the study represented a real-world survey of approach to acute rhythm management of AFl.

The study flow chart is presented in Figure 1. The mandatory inclusion criteria were an AFl or AF (I48 code in ICD) diagnosis based on the following ECG criteria: regular atrial activity at 250–350 bpm; “saw-tooth” pattern of inverted flutter waves in leads II and III; aVF or lack of P waves and absolute irregularity of rhythm; and indication for acute rhythm management based on the attending physician’s judgement.

The exclusion criteria were a lack of data concerning the conversion to the sinus rhythm, permanent AF or AFl, sick sinus syndrome or AF/AFl with bradycardia, age < 18 years, contraindication to acute rhythm control due to a lack of adequate anticoagulation if AF/AFl episode > 48 h, and chronic antiarrhythmic drug therapy (Figure 1).

The study was exempt from consent by the Ethics Committee of the Medical University of Silesia because of the retrospective registry-based approach. All data were gathered anonymously by review of electronic health records.

### 2.1. Efficacy and Safety end Points

The primary end point of the case-control study was successful AF termination by pharmacological cardioversion reflected by return to SR confirmed on a 12-lead electrocardiogram in the ED or cardiology department.

The composite safety end point was the occurrence of any serious adverse event following the administration of antiarrhythmic drug, including hypotension (decrease in systolic blood pressure of >40 mm Hg), bradycardia below 45 bpm, syncope, stroke or death.

### 2.2. Statistical Analysis

Statistical analysis was calculated using SPSS v.25.0 software (IBM Corp, Armonk, NY, USA). All continuous variables were submitted to a Shapiro–Wilk’s test and expressed as the median and 1–3 quartile range or mean and standard deviation (SD). All qualitative parameters are expressed as a crude number and percentage. The inter-group differences were compared using the Mann–Whitney U test. The relative risk (RR) ratio with 95% confidence interval (CI) limits were established for the efficacy of the amiodarone for pharmacological cardioversion of AFl and AF. In the nested case-control population, a univariate analysis of the different predictors of successful pharmacological cardioversion were delineated. All variables with *p* < 0.1 were included in a logistic regression analysis of the independent predictors of successful pharmacological cardioversion. A *p*-value < 0.05 was regarded as statistically significant throughout the analyses.

## 3. Results

### 3.1. Demographic and Clinical Characteristics of AFl Patients

During the registry duration, 727 were enrolled of whom 222 had AFl (30.5%) and 505 AF (69.5%) during the index consult in the emergency department. The data on the demographic and clinical characteristics are summarized in Table 1. The population of AFl was characterized by a median CHA2DS2-VASc score of 3 (2, 4) and median duration of AFl episode of 72 (16, 120) h. In the AFl population, former episodes of AF were documented in 72 patients (32.4%).

Different strategies of AFl management are summarized in Table 2. Catheter ablation was the most common treatment strategy in general (n = 90, 40.5%) and ablation ad hoc was applied in 54 patients (24.32%) while delayed in 36 patients (16.2%). Slightly fewer patients were treated with pharmacological cardioversion (PC) (n = 80, 36.0%) and electrical cardioversion (EC) (n = 74, 33.3%). The spontaneous return of sinus rhythm was reported in 21 patients (9.5%). The return of sinus rhythm was much more frequent after EC than PC (97.26% vs. 45.68%, *p* < 0.001).

### 3.2. Comparison of AFl vs. AF Cohort

Different demographic and clinical characteristics in AFl and AF patients are presented in Table 1. The aFl population was characterized by a higher proportion of men (65.3% vs. 46.3%, *p* < 0.001), required a longer hospitalization time, had a higher prevalence of diabetes mellitus (31.2% vs. 17.6%, *p* < 0.001), history of stroke/transient ischemic attack (10.3% vs. 4.4%, *p* = 0.003), had lower LVEF (47.2% vs. 54.7%, *p* < 0.001), greater left atrial diameter (43.4 vs. 42.3 mm, *p* = 0.017) and lower estimated glomerular filtration rate (68.0 vs. 71.5%, *p* = 0.023). Of note, patients with AFl were less frequently submitted to pharmacological cardioversion and more frequently to electrical cardioversion. The acute attempt at conversion of rhythm was generally less successful in AFl than AF (81.1% vs. 88.5%, *p* = 0.002).

### 3.3. Clinical Variables by Termination of AFl

The characteristics of the different variables depending on the successful termination of AFl are highlighted in Appendix A. The AFl patients with an unsuccessful attempt at rhythm conversion using different means had significantly lower LVEF compared to patients with return of sinus rhythm (44.4 ± 11.9 vs. 47.8 ± 12.4, *p* = 0.042). Other variables were comparable in both groups of patients.

### 3.4. Pharmacological Cardioversion with Amiodarone in AFl vs. AF: A Nested Case-Control Study

In the nested case-control study, 59 patients with AFl subject to PC with amiodarone matched in terms of age and sex with a control group of 60 patients with AF. Data on the demographic and clinical characteristics in both groups are summarized in Table 2. Both groups were comparable in terms of sex and age, EHRA class, CHA2DS2-VASc score and use of oral anticoagulation (*p* = 0.91). The patients with AFl had lower LVEF (*p* < 0.001), lower estimated glomerular filtration rate (*p* = 0.022) and higher troponin level (*p* = 0.004) than AF patients. Metoprolol or other betablockers were used more frequently in AFL than AF patients (55.9% vs. 35.0%, *p* < 0.001). The patients with AFl had a shorter duration of arrhythmia episode than AF counterparts (Table 2).

The efficacy of PC with amiodarone was significantly lower in the AFl than the AF group (39% vs. 65%, relative risk (RR) 0.60, 95% confidence interval (Cl): 0.41–0.87, *p* = 0.007). Still, the electrical cardioversion was equally efficacious in the AFl and AF cohort (95.7% vs. 100%, *p* = 0.976).

### 3.5. Univariate and Multivariate Predictors of Successful Pharmacological Cardioversion

The univariate analysis showed that successful pharmacological cardioversion was predicted by the type of arrhythmia (AFl—OR 0.34; 95%CI: 0.163–0.725, *p* = 0.005) and no use of intravenous metoprolol (OR 0.39; 95%CI: 0.173–0.892, *p* = 0.026). Presence of coronary artery disease (*p* = 0.85), arterial hypertension (*p* = 0.37), diabetes mellitus (*p* = 0.44), CHA2DS2-VASc score (*p* = 0.63), duration of arrhythmia episode (*p* = 0.38), estimated glomerular filtration rate (*p* = 0.34), hemoglobin level (*p* = 0.31), heart rate (*p* = 0.70), potassium level (*p* = 0.10), intravenous potassium supplementation (*p* = 0.23), EHRA class (*p* = 0.48), left atrial diameter (*p* = 0.23), LVEF (*p* = 0.15), male sex (*p* = 0.14), troponin level (*p* = 0.99), history of stroke (*p* = 0.45), white blood cell count (*p* = 0.09) and age (*p* = 0.66) failed to be associated with successful return of sinus rhythm.

In line with the results of logistic regression analysis, only presence of AFl was negatively associated with return of sinus rhythm (OR = 0.22, 95%CI: 0.091–0.529, *p* = 0.001). The area under the curve for the model was 0.671 (95%CI: 0.569–0.763, Hosmer–Lemeshow test, *p* = 1.0).

## 4. Discussion

The present retrospective analysis of AFl and AF patients consulted in an emergency department showed that the population of AFl patients was fraught with greater prevalence of comorbidities, including diabetes mellitus and chronic kidney disease, as well as a higher rate of impaired left ventricular systolic function and ischemic stroke in anamnesis and longer hospitalization time. Still, patients with AFl had a similar prevalence of coronary artery disease and arterial hypertension. Upon admission to the emergency department due to an acute episode of arrhythmia, 36% of the AFl and 89.5% of the AF patients were initially managed with pharmacological cardioversion (*p* < 0.001), while pre-emptive electrical cardioversion was utilized more often in the AFl cohort. The nested case-control substudy confirmed the widely known notion that AFl is refractory to acute rhythm management with AAD, as pharmacological cardioversion with amiodarone was successful only in 39% of the AFl patients vs. 65% of the AF patients (Figure 2). Moreover, the study found that AFl was an independent predictor of the efficacy of pharmacological cardioversion, while in the AFl cohort failure of termination of arrhythmia was linked to impaired LVEF.

The main findings of our trial are in line with a study by Cacioppo et al., who found that cardioversion with intravenous amiodarone led to the return of sinus rhythm in 31.6% of patients (n = 6/19) [20]. In this study, amiodarone was far less efficacious than IV ibutilide (90%, n = 18/20; OR 19.5); however, it was characterized by a relatively low study count [20]. In our study, no adverse actions of amiodarone were reported, while in the abovementioned study, bradycardia was documented in two patients (3%). Our study did not incorporate ibutilide, which is currently unavailable in Poland and, thus, its efficacy cannot be compared between trials. Both ibutilide and dofetilide were found to be particularly effective in the setting of AFl, but their use confers the risk of QTc interval prolongation and polymorphic ventricular tachycardia induction in 4% of cases [21]. The partial explanation of the low efficacy of pharmacological cardioversion in AFl cohort in comparison to the AF group might be related with the much lower rate of IV potassium supplementation (Table 2). Combined potassium and magnesium administration was associated with a higher rate of spontaneous return of sinus rhythm in AF, but not in the AFl population [22].

In the LADIP trial [23], the strategy of initial electrical cardioversion and prolonged oral amiodarone was compared with the upstream catheter ablation of AFl. The study showed that the strategy of direct referral for catheter ablation of AFl was associated with a much lower risk of arrhythmia recurrence (3.8% vs. 29.5%, *p* < 0.0001) [23]. For this reason, regardless of the initial mode of rhythm control management (electrical or pharmacological cardioversion), catheter ablation should be planned to prevent future arrhythmia recurrence [12]. The cited research providence evidence that provides additional information regarding the suboptimal efficacy of chronic amiodarone use for the prevention of AFl recurrence [23]. In our study, 24.3% of patients with AFl proceeded directly to ablation, while in 16.2% of patients the procedure was postponed and scheduled for subsequent hospitalization following initial rate and rhythm control management. These data deliver evidence of a suboptimal approach to AFl management, presumably deriving from deficiencies in the funding of the health care system.

Our study also provides data on the prevalence of coronary artery disease in the AFl population, contrary to the formerly published study by Iden et al., showing a five-fold increased risk of significant coronary artery stenosis in patients with AFl [8]. In our study, the prevalence of coronary artery disease was comparable in the AFl and AF groups, and the explanation for this discrepancy remains unknown. However, the result of the current research sheds light on the significantly greater prevalence of systolic dysfunction in the AFl population. Heart failure coexisting with AFl may be conditioned by tachycardia-induced cardiomyopathy, or AFl may result from macro-reentry circuit resulting from atrial fibrosis. Regardless of the pathophysiology, this observation should warrant prompt catheter ablation, which may lead to significant and prolonged improvement of LVEF [24].

It is vital to note that the prevalence of stroke or transient ischemic attack was significantly higher in the AFl than AF patients (Table 1). This finding is of paramount importance for stroke prevention and chronic anticoagulation. As there is a dispute over the risk of thrombus formation in patients with typical AFl, this observation together with the high level of coexistence of AFl together with AF (32.4%) warrants the institution of chronic oral anticoagulation in the AFl population, which is consistent with current European guidelines [6]. The question remains whether patients with AFl should continue anticoagulation following successful catheter ablation of AFl. High yearly risk of AF onset should prompt active screening for AF in this population [3].

## 5. Study Limitations

The main limitation of this study is its retrospective, registry-based design. Thus, the rate of complications of pharmacological cardioversion might have been underreported. Although 32.4% of patients consulted for AFl had a former history of AF episodes, the rate of conversion of AFl and AF during index hospitalization was not studied. Given the data’s incompleteness, the differentiation between typical and atypical flutter was impossible. The number of patients who were submitted to pharmacological cardioversion with amiodarone in the nested subanalysis was low. The groups receiving amiodarone were matched in terms of age and gender, but propensity score matching was not performed given the intrinsic difference between both arrhythmias, which would have a negative impact on the interpretation of amiodarone’s efficacy in both cohorts. Information on whether the current arrhythmic episode was the first in life or recurrent was unavailable. The study did not cover the time to conversion of rhythm in patients subject to pharmacological cardioversion.

## 6. Conclusions

In comparison to AF, AFl is associated with more pronounced cardiovascular risk reflected by higher prevalence of comorbidities. The efficacy of pharmacological cardioversion of AFl with amiodarone is low; hence, patients should initially be managed with electrical cardioversion and promptly referred for catheter ablation.

## Figures and Tables

**Figure 1 jcm-12-04262-f001:**
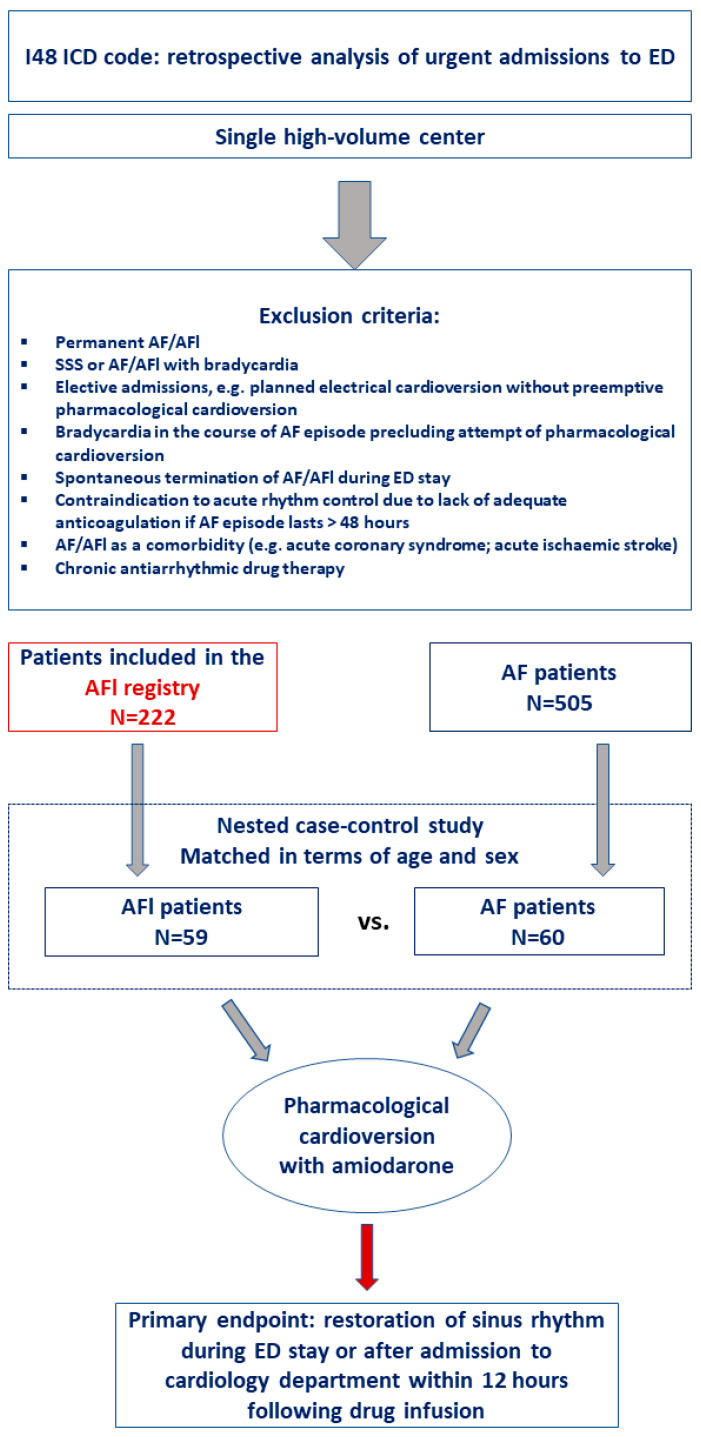
Study flow chart. AF—atrial fibrillation; AF—atrial flutter; ED—emergency department; SSS—sick sinus rhythm; ICD—international classification of diseases.

**Figure 2 jcm-12-04262-f002:**
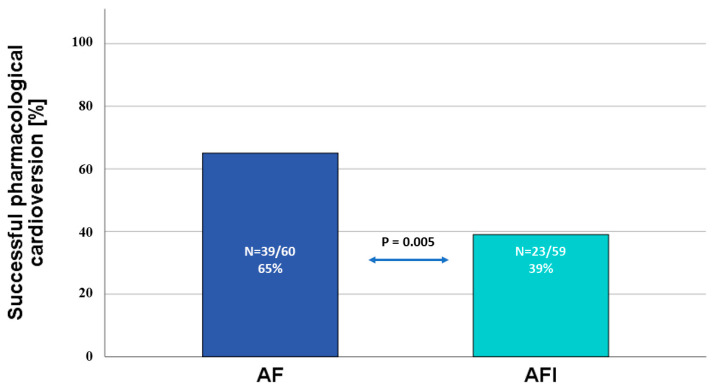
Rate of successful pharmacological cardioversion and AF and AFl. AF—atrial fibrillation; AFl—atrial flutter.

**Table 1 jcm-12-04262-t001:** Baseline characteristics of the study population depending on the type of arrhythmia: AFl vs. AF.

Variable	AFl—Whole Population N = 222	AF—Whole Population N = 505	*p*-Value
Male sex	145 (65.32%)	234 (46.34%)	<0.001
Age, years	67.3 ± 11.6	65.63 ± 11.9	0.130
Hospitalization time, days	3.2 ± 2.4	1.7 ± 1.3	<0.001
Duration of arrhythmic episode, hours	72 (16; 120)	10.5 (5; 24)	<0.001
Heart rate, bpm	115.1 ± 38.5	120.8 ± 24.7	0.150
Heart rate > 130 bpm	75 (36.4%)	183 (39.1%)	0.502
EHRA class	2 (2; 3)	3 (2; 3)	0.820
CHA_2_DS_2_-VASc, pts	3 (2; 4)	3 (2; 4)	0.056
Arterial hypertension	151 (74.4%)	363 (71.9%)	0.499
Diabetes mellitus	63 (31.2%)	89 (17.6%)	<0.001
CAD/PAD	68 (33.2%)	159 (31.5%)	0.663
Former TIA/stroke	21 (10.3%)	22 (4.4%)	0.003
History of PVI	20 (9.8%)	32 (6.3%)	0.197
LVEF, %	47.2 ± 12.4	54.71 ± 8.3	<0.001
LVEF < 50%	72 (38.7%)	68 (13.5%)	<0.001
LAd, mm	43.4 ± 6.1	42.25 ± 5.16	0.017
LAd > 40 mm	139 (77.7%)	275 (75.1%)	0.518
TnT > 0.014 pg/mL	63 (54.3%)	116 (34.5%)	<0.001
SCr, mg/dL	1.1 ± 0.8	0.99 ± 0.33	<0.001
eGFR, mL/min	68 ± 18.5	71.5 ± 17.3	0.023
eGFR < 60 mL/min per 1.73 m^2^	62 (30.7%)	113 (22.38%)	0.021
Potassium level, mEq/L	4.37 ± 0.4	4.26 ± 0.4	0.010
WBC × 1000/μL	8.1 ± 2.9	7.81 ± 2.85	0.226
Hemoglobin, g/dL	14.0 ± 1.9	14.3 ± 1.5	0.134
TSH, mIU/L	1.87 (1.09; 3.06)	1.84 (1.10; 2.71)	0.913
Initial pharmacological cardioversion	80 (36.0%)	452 (89.5%)	<0.001
Electrical cardioversion	74 (33.3%)	53 (10.5%)	<0.001
Successful termination of arrhythmia until discharge	180 (81.1%)	447 (88.5%)	0.002

AFl—atrial flutter; AF—atrial fibrillation; bpm—beats per minute; eGFR—estimated glomerular filtration rate; EHRA—European Heart Rhythm Association; CAD—coronary artery disease; LAd—left atrial diameter; LVEF—left ventricular ejection fraction; PAD—peripheral artery disease; TIA—transient ischemic attack; PVI—pulmonary vein isolation; TnT—troponin T; SCr—serum creatinine concentration; WBC—white blood cell count; TSH—thyroid-stimulating hormone.

**Table 2 jcm-12-04262-t002:** Case-control comparison between AFl and AF in terms of clinical variables and efficacy of pharmacological cardioversion with amiodarone.

Variable	AF N = 60	AFl N = 59	*p*-Value
Demographic Characteristics			
Male sex	32 (53.3%)	39 (66.1%)	0.156
Age, years	69 (57.5–74)	68 (61–75)	0.527
Hospital admission	23 (38.3%)	49 (83.1%)	<0.001
Hospitalization time, days	1.7 ± 1	4.3 ± 3.3	<0.001
Comorbidities			
Arterial hypertension	44 (73.3%)	44 (74.6%)	0.877
Diabetes mellitus	13 (21.7%)	18 (30.5%)	0.272
Heart rate > 130 bpm	22 (37.3%)	15 (27.3%)	0.254
Vascular Disease	17 (28.3%)	19 (34.6%)	0.473
History of ischemic stroke/TIA	4 (6.7%)	2 (3.6%)	0.456
History of PVI	3 (5%)	6 (10.7%)	0.250
Anticoagulation use	36 (60%)	36 (61%)	0.910
Metoprolol or other betablocker use	21 (35%)	33 (55.9%)	<0.001
Arrhythmia Characteristics			
EHRA	3 (2; 3)	3 (2; 3)	0.373
CHA_2_DS_2_-VASc score	3 (1; 4)	3 (2; 5)	0.381
Heart rate, bpm	122.4 ± 23.1	111.5 ± 39.4	0.121
Duration of episode, hours	12 (7; 24)	24 (12; 72)	0.035
Duration ≥ 48 h	3 (5.0%)	19 (52.8%)	<0.001
Duration ≥ 7 days	1 (1.7%)	17 (43.6%)	<0.001
Echocardiographic Parameters			
LVEF, %	54.3 ± 9.6	45.1 ± 13.6	<0.001
LVEF < 50%	7 (11.7%)	21 (42.0%)	<0.001
LAd, mm	41.4 ± 4.4	45 ± 5.3	<0.001
LAd ≥ 40 mm	37 (77.1%)	38 (82.6%)	0.505
Laboratory Tests			
Troponin ≥ 0.014	14 (34.2%)	18 (62.1%)	0.021
eGFR, mL/min	73.2 ± 15.4	64.1 ± 20.9	0.022
eGFR ≤ 60 mL/min	11 (20.4%)	17 (32.7%)	0.150
Potassium level, mEq/L	4.3 ± 0.5	4.5 ± 0.5	0.224
SCr, mg/dL	1.0 ± 0.2	1.4 ± 1.5	0.040
Troponin T, ng/mL	0.01 ± 0.01	0.04 ± 0.07	0.004
WBC × 1000/mm^3^	7.7 ± 2.5	9 ± 3.5	0.036
Hemoglobin, g/dL	14.3 ± 1.7	14.1 ± 2	0.645
TSH, uIU/mL	2.2 ± 2.3	2.4 ± 1.8	0.378
Treatment			
Amiodarone dose, mg	300 (150; 450)	300 (150; 450)	0.662
Transesophageal echocardiography	5 (8.3%)	16 (27.1%)	0.007
Potassium i.v.	42 (70%)	19 (32.2%)	<0.001
Successful PC	39 (65%)	23 (39%)	0.005
EC	7 (11.7%)	23 (39%)	0.001
Successful EC	7 (100%)	22 (95.7%)	0.976
Return to sinus rhythm	46 (76.7%)	45 (76.3%)	0.867
Composite safety endpoint	0	0	-

AF—atrial fibrillation; AFl—atrial flutter; bpm—beats per minute; CAD—coronary artery disease; eGFR—estimated glomerular filtration rate; EC—electrical cardioversion; EHRA—European Heart Rhythm Association; LAd—left atrial diameter; LVEF—left ventricular ejection fraction; PAD—peripheral artery disease; TIA—transient ischemic attack; PC—pharmacological cardioversion; EC—electrical cardioversion; PVI—pulmonary vein isolation; TnT—troponin T; SCr—serum creatinine concentration; WBC—white blood cell count; TSH—thyroid-stimulating hormone.

## Data Availability

The source data of the article is available upon request.

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
