# Peer review of "Clinical Characteristics of Atrial Flutter and Its Response to Pharmacological Cardioversion with Amiodarone in Comparison to Atrial Fibrillation"

_jcm, 2023, doi:10.3390/jcm12134262_

Round 1

Reviewer 1 Report

Wybraniec et al describes “Clinical characteristics of atrial flutter and its response to phar-1 macological cardioversion with amiodarone in comparison to 2 atrial fibrillation”. Article is reasonably well written, however there are some concerns.

1.       There is no clarity regarding the dose of amiodarone that was usually used in the study. How long was amiodarone administered. Was a specific protocol used?

2.       Tables – table 2 – use Aflutter and atrial fibrillation short forms consistently across the manuscript. If using full forms, use that consistently across the manuscript.

3.       Table 2  mentions electrical cardioversion – was electrical cardioversion performed after the 12 hr period was completed in patients who were not successfully converted by amiodarone. Please clarify where electrical cardioversion comes in the algorithm used.

4.       Only 76% of patients appeared to get back to sinus rhythm, is this after attempted cardioversion with electrical means also? Please clarify. Also clarify the time duration needed to get back in sinus rhythm.

5.       Prior history of afib flutter needs to be added in the baseline characteristics.

6.       How was duration of afib or flutter determined for the purpose of the study .

minor english language editing is needed. 

Author Response

Reviewer #1

We would like to thank the Reviewer for all constructive remarks. We have made an effort to address all the issues raised by the Reviewer.

Wybraniec et al describes “Clinical characteristics of atrial flutter and its response to pharmacological cardioversion with amiodarone in comparison to 2 atrial fibrillation”. Article is reasonably well written, however there are some concerns.

  1. There is no clarity regarding the dose of amiodarone that was usually used in the study. How long was amiodarone administered. Was a specific protocol used?

The study represents a retrospective analysis, thus no specific protocol for amiodarone infusion was used. The decision regarding amiodarone use was at the discretion of attending physician. The median dose of amiodarone was presented in Table 2 and it was comparable in AF and AFl group (300 mg vs. 300 mg, p=0.662). The methodology of amiodarone infusion was described in method’s section and involved diverse dose and rates of amiodarone dissolved in 5% glucose solution. In general, the majority of patients received 300 mg amiodarone dissolved in 50 ml of 5% glucose administered over 2 h, while some patients received preemptive bolus of 150 amiodarone. We have added the last sentence to the methods section.

  1. Tables – table 2 – use Aflutter and atrial fibrillation short forms consistently across the manuscript. If using full forms, use that consistently across the manuscript.

We are grateful for this suggestion. We have modified the full forms of AF and AFl.

  1. Table 2  mentions electrical cardioversion – was electrical cardioversion performed after the 12 hr period was completed in patients who were not successfully converted by amiodarone. Please clarify where electrical cardioversion comes in the algorithm used.

Electrical cardioversion was performed after 12 hour observation period according to the clinician’s judgement based on the chance of sinus rhythm restoration if pharmacological cardioversion failed to restore sinus rhythm. In patients with unsuccessful pharmacological cardioversion and no attempt at electrical cardioversion, rate control strategy was introduced with delayed elective electrical cardioversion. We have added this information to the methods section for clarity.

  1. Only 76% of patients appeared to get back to sinus rhythm, is this after attempted cardioversion with electrical means also? Please clarify. Also clarify the time duration needed to get back in sinus rhythm.

In Table 2, similar proportion of patients regained sinus rhythm (76.7% in AF and 76.3% in AFl group, p=0.867) following both pharmacological and electrical cardioversion, however, if pharmacological cardioversion failed, not all patients were subject to electrical cardioversion. The efficacy of electrical cardioversion was 100% in AF group (n=7 / 7) and 95.7% in AFl group (n=22 / 23). 13 AFl patients with unsuccessful PC and no attempt at electrical cardioversion were scheduled for elective EC or ablation.

The study did not cover the time to conversion of rhythm in patients subject to pharmacological cardioversion and this information has already been included in limitations section.

  1. Prior history of afib flutter needs to be added in the baseline characteristics.

Regrettably we have not included the information on whether the current episode of arrhythmia was the first in life or recurrent. We therefore are unable to include this information in the baseline characteristics. We have added this information to the limitations section.

  1. How was duration of afib or flutter determined for the purpose of the study .

The duration of AF or AFl episode was determined based on the interview with patients, meaning that the onset of symptoms was used as the beginning of the arrhythmia onset.

Reviewer 2 Report

Dear Sir/Madam,

I had the opportunity to act as a reviewer on the recent submission by Wybraniec et al. to the Journal of Clinical Medicine.

The authors present an interesting study analyzing the efficacy of pharmacologic cardioversion with amiodarone of atrial fibrillation versus atrial flutter in a high-volume emergency department. The main finding was that the pharmacologic cardioversion was higher in the atrial fibrillation versus atrial flutter patients.

The manuscript is very well structured and written. However, some issues need to be addressed: 

1.     How do the authors explain, that the patients with atrial flutter were sicker than the patients with atrial fibrillation? This is somehow discrepant to the daily routine, as atrial fibrillation patients exhibit more comorbidities.

2.     On what basis was the decision taken to compare 60 patients in each group?

3.     Why were there more patients with stroke in the atrial flutter group? Is there a plausible explanation for this finding?

4.     Line 119: why was the cutoff of 0.1 chosen and not 0.05 as usual?

Best regards,

Author Response

Reviewer #2

We would like to thank the Reviewer for the effort of reviewing our manuscript and all the constructive suggestions.

Dear Sir/Madam,

I had the opportunity to act as a reviewer on the recent submission by Wybraniec et al. to the Journal of Clinical Medicine.

The authors present an interesting study analyzing the efficacy of pharmacologic cardioversion with amiodarone of atrial fibrillation versus atrial flutter in a high-volume emergency department. The main finding was that the pharmacologic cardioversion was higher in the atrial fibrillation versus atrial flutter patients.

The manuscript is very well structured and written. However, some issues need to be addressed: 

  1. How do the authors explain, that the patients with atrial flutter were sicker than the patients with atrial fibrillation? This is somehow discrepant to the daily routine, as atrial fibrillation patients exhibit more comorbidities.

We thank the Reviewer for this opinion. According to literature and our experience atrial flutter, including typical and atypical form, is associated with structural heart disease more than atrial fibrillation is. Although comorbidities like arterial hypertension or diabetes mellitus are risk factors of left atrial enlargement, fibrosis and create substrate for atrial fibrillation, the presence of atrial flutter is a marker of diseased heart with frequent presence of heart failure, occult coronary artery disease, fibrotic scars following cardiac surgery, significant chronic respiratory failure, alcohol abuse. Our results are somewhat in line with this data. Beneath we enclose studies that support this statement:

  • Iden L, Richardt G, Weinert R, Groschke S, Toelg R, Borlich M. Typical atrial flutter but not fibrillation predicts coronary artery disease in formerly healthy patients. Europace 2021; 23:1227-1236.
  • Granada J, Uribe W, Chyou PH, Maassen K, Vierkant R, Smith PN, Hayes J, Eaker E, Vidaillet H. Incidence and predictors of atrial flutter in the general population. J Am Coll Cardiol. 2000 Dec;36(7):2242-6.
  • Halligan SC, Gersh BJ, Brown RD, Rosales AG, Munger TM, Shen WK, Hammill SC, Friedman PA. The natural history of lone atrial flutter. Ann Intern Med. 2004 Feb 17;140(4):265-8.
  • Compagnucci P, Casella M, Bagliani G, Capestro A, Volpato G, Valeri Y, Cipolletta L, Parisi Q, Molini S, Misiani A, Russo AD. Atrial Flutter in Particular Patient Populations. Card Electrophysiol Clin 2022; 14:517-532.

  1. On what basis was the decision taken to compare 60 patients in each group?

In our real-world database of patients subject to pharmacological cardioversion, we found 59 consecutive patients with AFl subject to amiodarone. We wanted to compare its efficacy with atrial fibrillation so we selected random 60 patients with AF submitted to pharmacological cardioversion with amiodarone, which were matched in terms of age and sex.

  1. Why were there more patients with stroke in the atrial flutter group? Is there a plausible explanation for this finding?

As atrial flutter represents a more organized arrhythmia, the risk of thrombus formation withing left atrial appendage should be smaller. Still the hitherto data showed similar risk of systemic thromboembolism in AFl and AF, suggesting the need for similar antithrombotic treatment regimen in both arrhythmias. We cannot provide a plausible explanation for this observation, but this finding underscores the need for anticoagulation in these subset of patients.

  1. Line 119: why was the cutoff of 0.1 chosen and not 0.05 as usual?

The p-value cut-off of <0.1 was used to identify variables, which were included in logistic regression model. The p-value cut-off of <0.05 was regarded as statistically significant throughout the analyses – we have included this information in the methods section.